# Results of a Continuous Quality Improvement Initiative of the *Contemporaneous Model* of Service Delivery

**DOI:** 10.3390/geriatrics7050118

**Published:** 2022-10-20

**Authors:** Atul Sunny Luthra, Adam Millar

**Affiliations:** 1Department of Psychiatry and Behavioral Neuroscience, McMaster University, Hamilton, ON L8M1W9, Canada; 2Schlegel Research Institute in Aging, Waterloo, ON N2J0E2, Canada; 3Faculty of Health Science, McMaster University, Hamilton, ON L8N3Z5, Canada

**Keywords:** gero-psychiatry assessment team, *Contemporaneous Model*, emergency department, specialized dementia behavioral service

## Abstract

The *Contemporaneous Model* of service delivery serves to manage behavioral expression in residents of long-term care homes with a diagnosis of advanced neurocognitive disorder. Its effectiveness is benchmarked in preventing the residents, on its active caseload, from seeking assistance in the emergency department and the dementia behavioral inpatient units for behavioral risks. The results of the three years of operation of the *Contemporaneous Model* of service delivery, for the years 2017–2018, 2018–2019, and 2019–2020, are presented here. These results are supportive of this model of service delivery as an effective way to reduce the burden of patients with advanced neurocognitive disorder with behavioral expressions on the emergency departments and specialized dementia behavioral services. It has the potential for becoming the gold standard model of service delivery in the Canadian health care system.

## 1. Introduction

Comprehensive geriatric assessment teams (CGAT) came into existence in the late 1990′s with the focus of attending to the medical and the physical needs of the older persons, initially on the medical and the surgical units of the acute care hospitals, and subsequently in the community. A meta-analysis of the outcome of the operations of the comprehensive geriatric assessment teams (CGAT) revealed that only the teams that had ‘adequate integration with the referral teams, controlled medication prescribing and provided extended ambulatory follow-ups were effective in reducing mortality [1,2], lengths of stay, improving functioning and reducing *re-admission rates’* [2,3,4]. Gero-psychiatry assessment teams (G-PAT) have not evolved on the inpatient medical and surgical units of the acute care hospitals, as this role has been fulfilled by Medical Psychiatry services [5,6]. Gero-psychiatry assessment teams have evolved in the community, in both non-institutional and institutional settings. Several models of geriatric mental health outreach teams are in operation in the non-institutional settings. These include partnerships between specialized geriatric services and primary care physicians [7], tertiary care-based multidisciplinary services [8], nurse-lead outreach teams [9], and social-worker lead multidisciplinary teams [10]. The model of outreach services in the institutional settings include psychiatrist-centered models, multidisciplinary team model with a psychiatry lead, and social worker and nurse lead teams [11]. Literature on the operations of the geriatric mental health outreach teams in Canada is very limited, though anecdotal information revels ‘consultation model’ with limited follow-ups [12].

The Geriatric Psychiatry and Medicine services have been in operation at Hamilton Health Sciences (HHS), Hamilton, Ontario, Canada, since the early 1990′s. The operations of the geriatric psychiatry assessment team (G-PAT) at Hamilton Health Sciences, Hamilton, Ontario, Canada, were reorganized in 2013 in accordance with the position paper put forth by the Ministry of Health and Regional Geriatric Program, with an aim for reducing the burden of residents in long-term care homes (LTCH) on emergency departments [4,13]. This new model of service delivery was called the *Contemporaneous Model* [13].

The results of the pilot project on the qualitative and the quantitative evaluation of the *Contemporaneous Model* of service delivery of G-PAT at HHS were published in 2015 [13]. The results of the quantitative evaluation of G-PAT revealed that 91% of the patients on its active caseload were prevented from visiting the emergency departments to seek help for behavioral risks [13,14]. In 2016, the *Contemporaneous Model* of service delivery incorporated an innovative model of clinical approach (LuBAIR™ approach) for assessing and managing behavioral expressions in advanced neurocognitive disorder [15,16,17,18].

Due to the changing demographics of residents of the LTCH, age criteria (65 and over) were eliminated and the focus of G-PAT changed to assessment and management of behavioral expressions in residents of LTCH who have *cognitively complex issues, with or without concurrent mental illness, regardless of the age or etiology* [19,20,21,22,23,24]. Once all of the above aforementioned changes to G-PAT were fully operationalized in 2016, it allowed for the establishment of the next phase of the evaluative framework under the governance of *Continuous Quality Improvement.*

Patients with the diagnosis of *cognitively complex issues, with or without concurrent mental illness, regardless of age or etiology,* present a significant burden on the emergency departments of the acute care hospitals in the Canadian healthcare system [25,26]. Emergency departments have become the ‘point of entry’ for the patients with complex cognitive and medical illnesses into the general internal medicine units of the acute care hospitals [14,25,26]. Once admitted, the flow of this cohort into specialized inpatient services or back into the community is impeded by the lack of resources to support their clinical needs. This invariably results in a significant reduction in the capacity of the system to manage the patients, which should typically fall under its purview and include acute onset medical illnesses or acute exacerbation of chronic medical illnesses [27,28,29,30]. The reduced capacity of the general internal medicine units, due to occupancy from the cognitively complex patients with high behavioral and daily care risks, has been identified as one of the contributory variables to the emergence of *Hallway Medicine* [31,32,33]. Hence, the prevention of this cohort of the population from presenting to the emergency departments, or reducing the use of specialized inpatient units, is one of the ways of enhancing the capacity of the system to cope with the increasing clinical care needs of the aging population [31,32,33].

The operations of the G-PAT, under the governance of the *Contemporaneous Model* of service delivery, are positioned within the healthcare system to address the clinical care needs of this cohort of the population, and its success is benchmarked in reducing the burden of this cohort of the population on the emergency departments and specialized inpatient services. This evaluative framework study was conducted under the governance of the *Continuous Quality Improvement Initiative (Edwards, 2008).* The results of the effectiveness of the operations of G-PAT for the fiscal years 2017–2018, 2018–2019, and 2019–2020, are presented in this manuscript.

## 2. Methods

### 2.1. Design

The G-PAT consists of the lead geriatric psychiatrist and four case managers. Three of the case managers are occupational therapists by training and one is a registered nurse. This project was established based on an evaluative framework. The principles governing the project design were derived from the ‘observation case study’ design. Each individual case on the active caseload of the G-PAT was reviewed on the basis of their need to visit the emergency department or requiring a referral to a specialized behavioral unit for behavioral risks. This was done for each case on the active caseload of G-PAT over three fiscal years of 2017–2018, 2018–2019, and 2019–2020.

### 2.2. Participants

There are 27 long-term care homes in the City of Hamilton, Ontario, Canada. The admission to LTCH is determined by an independent health agency, based upon their evaluation framework. This framework is based on the resident requiring total assistance with basic activities of daily care, regardless of whether they have cognitive impairment. However, the vast majority of the residents admitted to LTCH suffer from varying degrees of cognitive impairment [25,33]. Age, as a criterion for exclusion, has been replaced within the system by focusing on function, regardless of age, and medical or psychiatric diagnosis. The presence of behavioral expressions, which are associated with any degree of risk, is often an exclusion criterion for admission. All the residents with *Cognitively Complex* issues, which are referred to as G-PAT, experience an escalation of existing BE, and are now associated with risks or the emergence of new BE with associated risks. All of the residents with *Cognitively Complex* issues who develop BE with associated risks must be assessed by the in-house behavioral team, with input from Behavioral Supports Ontario [15,17,18,23] and/or psychogeriatric resource consultants [13,15] prior to qualifying for a referral to G-PAT. The most responsible physician must initiate the referral to G-PAT, after all of these inputs have failed to mitigate the risks associated with the BE.

This information for Figure 1 was obtained from the St. Joseph Hospital (SJH), Hamilton, Ontario, emergency department database. In Hamilton, there is a centralized intake process for all mental health patients which require an emergency mental health assessment. Individuals with advanced neuro-cognitive disorders who are exhibiting behavioral expressions are grouped under the mental health rubric, and, regardless of the place of origin (residential homes, retirement homes, or long-term care homes), are triaged through the psychiatric emergency services (PES) of SJH.

### 2.3. Interventions

Once the referral is initiated by the MRP, the intake office at the Centre for Healthy Aging reviews relevant case documentation and reaches out to the referral source and family for additional information, as needed, to establish the priority for the referral in accordance with the following criteria. The priority levels are governed by the definition of *severity* posited under the LuBAIR™ paradigm [13,17,22,23,24].

(a)Priority Level-1: No response of the behavioral expressions and associated risks to all of the interpersonal or environmental interventions, (IPI and EI), respectively.(b)Priority Level-2: Response of the behavioral expressions with associated risks to IPI and/or EI, only for them to relapse once IPI and EI are withdrawn (un-sustained response).(c)Priority Level-3: Response of behavioral expressions and associated risks to IPI and/or EI, which persists even after they are withdrawn (sustained response).

Once prioritized, residents are scheduled, and the case is assigned to one of the four case managers. The case manager will gather information, from all available sources including the family, in preparation for a combined visit with the geriatric psychiatrist. Subsequent follow-ups can take the form of telephone contacts, solo visits by the case manager, and scheduled follow up visits by the geriatric psychiatrist and the case manager. At each of these subsequent contacts, the impact of the changes in the medications on the individual’s functional abilities is evaluated as a measure of side effects from the medications, the role of inter-current medical or milieu contributors are determined, the changes in the frequency, duration, and severity of the identified quality of BE is established, along with the impact of the associated risks. Based upon this evaluation, further recommendations are implemented. This process is followed at each subsequent visit until the complete resolution of the risks, at which point the patient is discharged from the service.

### 2.4. Measurements

#### 2.4.1. Primary Outcome Measure

Total number of visits to the ED for the patients on the active caseload of G-PAT for behavioral risks.Total number of cases on the active caseload of G-PAT referred to the specialized dementia behavioral unit at the St. Peter’s Site of Hamilton Health Sciences.

#### 2.4.2. Secondary Outcome Measures

The total number of new consults and ‘unique’ cases for each fiscal year.Time to respond to the initial assessment of patients under priority level-1.Time to first follow-up on assessments done on priority level-1.Total number of doctor/CM visits in each fiscal year.Total number of CM solo visits in-between the doctor/CM visits.Total number of telephone contacts in-between doctor/CM and CM solo visits.

#### 2.4.3. Teriatry Outcome Measure

The total number of visits by the residents of LTCH in the City of Hamilton to the emergency department of St. Joseph Hospital, for all reasons (medical, surgical or behavioral), between December 2017 and June 2021 were collected. Also, the total number of visits by the residents of LTCH to the emergency department for behavioral reasons were separated.

### 2.5. Analysis

Descriptive statistics were used to establish the rates of desired outcomes, and subsequently evaluate the service. The health record of each of the patients on the active caseload of G-PAT was reviewed at the end of each fiscal year to establish if they were sent to the emergency department to seek assistance for behavioral risks. The health record of each patient on the active caseload of G-PAT was reviewed at the end of each fiscal year to determine if they required a visit to the emergency department for intervention. This number was totaled, and then expressed as a percentage of the total number of patients on the active caseload of G-PAT. Similarly, the health record of each patient on the active caseload of G-PAT was reviewed at the end of each fiscal year to determine if they required a referral to the specialized dementia behavioral unit for stabilization. This number was totaled and then expressed as a percentage of the total number of patients on the active caseload of G-PAT.

## 3. Results

### 3.1. 2017–2018. Table 1 Provides the Timelines for the Initial Assessment and First Follow-Up Visits for 2017/2018

#### 3.1.1. Primary Outcome


The total number of visits to the ED for the patients on the active caseload of G-PAT for behavioral reasons was 11. This accounted for 2.4% of the cases on the active caseload of G-PAT who required the assistance of ED for behavioral risks. G-PAT was able to prevent 97.6% of the cases on its active caseload from visiting the emergency department for behavioral risks.The total number of cases on the active caseload of G-PAT that was referred to the specialized behavioral unit at St. Peter’s Hospital was 16 out of a total active caseload of 460. This accounted for 3.4% of the cases on the active caseload of G-PAT that required the services of a specialized behavioral care unit. G-PAT was able to manage the behavioral risks in 96.6% of the cases on its active caseload.


#### 3.1.2. Secondary Outcomes

The total number of new consults for this fiscal year was 217. The total number of ‘unique’ cases on the active caseload for the fiscal year was 460.The ‘mean’ time to response to the initial assessment of the priority 1 referrals was 31.3 days. However, the range of the response times for priority 1 referrals was from 10.8 days to 68.3 days. The ‘median’ time of response to the initial assessment of the priority 1 referrals was 15 days.The ‘mean’ time until the first follow-up appointment for the priority 1 referrals was 24.6. The range of time until the first follow-up appointment for priority 1 referrals was from 16 to 37 days.The total number of ‘face-to-face’ follow-up visits by the geriatric psychiatrist and the assigned case manager for the patients on the active caseload of G-PAT was 1126.The total number of ‘solo’ follow-up visits by the case managers for the patients on the active caseload of G-PAT was 1039.The total number of ‘telephone’ reviews by the geriatric psychiatrist and the case manager of the patients on the active caseload of G-PAT was 1223.

**Table 1 geriatrics-07-00118-t001:** The timelines for the initial assessment and first follow-up visits for 2017/2018.

Appointment Description	2017–2018												
	17-Apr	May	Jun	Jul	Aug	Sep	Oct	Nov	Dec	18-Jan	Feb	Mar	Total
The total number of new consults	15	19	23	16	23	16	19	16	14	21	21	14	217
Acceptance to first appointment “priority 1” (mean) in days	14.0	11.8	14.5	30.3	52.8	29.2	68.3	53.8	27.0	24.9	10.8	27.0	31.3
Acceptance to first appointment “priority 1” (median) in days	14.5	8.0	14.5	25.0	14.0	21.0	24.0	43.0	8.0	28.0	14.0	18.0	15.0
First follow-up appointment for “priority 1” (mean) in days	23.0	20.3	37.0	27.8	16.7	19.0	21.0	34.0	32.0	22.1	16.0	24.0	24.6
Face-to-face follow-up visits with the doctor and the case manager	104	75	107	88	111	70	83	105	87	93	92	111	1126
Solo face-to-face case manager visits	86	98	104	101	69	68	79	78	85	95	75	101	1039
Telephone follow-ups by the doctor and/or case manager	113	111	148	98	88	107	70	97	92	82	110	107	1223
**Follow ups by the team (total)**	**303**	**284**	**359**	**287**	**268**	**245**	**232**	**280**	**264**	**270**	**277**	**319**	**3388**
The total active ‘unique’ cases	30	28	32	28	33	34	39	40	45	42	54	55	460
ED visits for behavioral issues	0	1	1	1	1	1	1	1	1	1	1	1	11 (2.4%)
Active case load of G-PAT referred to BH	1	2	1	2	1	2	1	1	2	0	1	2	16 (3.4%)

### 3.2. 2018–2019. Table 2 Shows the Timelines for the Initial Assessment and First Follow-Up Visits for 2018/2019

#### 3.2.1. Primary Outcome Measures


The total number of visits to the ED for the patients on the active caseload of G-PAT for behavioral reasons was seven. This accounted for 1.4% of the cases on the active caseload of G-PAT who required the assistance of ED for behavioral risks. G-PAT was able to prevent 98.6% of the patients on its active caseload from visiting the emergency department for behavioral risks.The total number of cases on the active caseload of G-PAT who were referred to the specialized behavioral unit at St. Peter’s Hospital was 11 out of a total active caseload of 513. This accounted for 2% of the cases on the active caseload of G-PAT that required the services of a specialized behavioral care unit. G-PAT was able to manage the behavioral risks in 98% of the cases on its active caseload.


#### 3.2.2. Secondary Outcome Measures

The total number of new consults for this fiscal year was 225. The total number of unique cases on the active caseload for this fiscal year was 513.The ‘mean’ time to response to the initial assessment of the priority 1 referrals was 24.6 days. However, the range of the response times for priority 1 referrals was from 14.7 days to 55.8 days. The ‘median’ time of response to the initial assessment of the priority 1 referrals was 19 days.The ‘mean’ time until the first follow-up appointment for the priority 1 referrals was 35.5. The range of time until the first follow-up appointment for priority 1 referrals was from 21 to 49 days.The total number of ‘face-to-face’ follow-up visits by the geriatric psychiatrist and the assigned case manager for the patients on the active caseload of G-PAT was 1093.The total number of ‘solo’ follow-up visits by the case managers for the patients on the active caseload of G-PAT was 735.The total number of ‘telephone’ reviews by the geriatric psychiatrist and the case manager of the patients on the active caseload of G-PAT was 1130.

**Table 2 geriatrics-07-00118-t002:** The timelines for the initial assessment and first follow-up visits for 2018/2019.

Appointment Description	2018–2019												
	18-Apr	May	Jun	Jul	Aug	Sep	Oct	Nov	Dec	19-Jan	Feb	Mar	Total
The total number of new consults	18	22	19	24	10	19	22	16	4	26	20	25	225
Acceptance to first appointment “priority 1”(mean) in days	55.8	15.7	16.0	15.1	22.6	41.7	14.7	30.0	42.0	23.2	19.3	20.5	24.6
Acceptance to first appointment “priority 1”(median) in days	29.0	13.0	14.5	15.0	13.0	49.0	10.0	29.0	48.0	16.0	18.5	20.5	19.0
First follow-up appointment “priority 1”(mean) in days	30.3	31.0	44.0	37.8	21.0	49.0	41.2	37.2	28.0	40.7	29.3	37.0	35.5
Face-to-face follow-up visits the doctor and the case manager	72	126	90	101	116	113	90	90	47	95	93	60	1093
Solo face-to-face case manager visits	76	77	89	81	85	74	80	13	33	33	54	40	735
Telephone follow-ups by the doctor and/or case manager	97.0	120.0	137.0	117.0	110.0	88.0	116.0	75.0	88.0	75.0	57.0	50.0	1130.0
**Follow ups by the team (total)**													**2958**
The total active caseload for the year	38	36	41	35	44	42	44	36	46	43	53	55	513
ED visits for behavioral issues	1	0	0	1	0	0	1	0	1	0	1	2	7 (1.4%)
Active case load of G-PAT referred to BH	0	1	1	1	1	1	1	1	1	1	1	1	11 (2%)

### 3.3. 2019–2020. Table 3 Provides the Timelines for the Initial Assessment and First Follow-Up Visits for 2019/2020

#### 3.3.1. Primary Outcome Measure


The total number of visits to the ED for the patients on the active caseload of G-PAT for behavioral reasons was four. This accounted for 0.8% of the cases on the active caseload of G-PAT that required the assistance of ED for behavioral risks. G-PAT was able to prevent 99.2% of the patients on its active caseload from visiting the emergency department for behavioral risks.The total number of cases on the active caseload of G-PAT that were referred to the specialized behavioral unit at St. Peter’s Hospital was seven out of a total active caseload of 525. This accounted for 1.4% of the cases on the active caseload of G-PAT that required the services of a specialized behavioral care unit. G-PAT was able to manage behavioral risks in 98.6% of the patients on its active caseload.


#### 3.3.2. Secondary Outcome Measure

The total number of new consults for this fiscal year was 213. The total number of unique cases on the active caseload for this fiscal year was 525.The ‘mean’ time to response to the initial assessment of the priority 1 referrals was 45.4 days. However, the range of the response times for priority 1 referrals was from 34.2 days to 59 days. The ‘median’ time of response to the initial assessment of the priority 1 referrals was 60 days.The ‘mean’ time until the first follow-up appointment for the priority 1 referrals was 42.2 days. The range of time until the first follow-up appointment for priority 1 referrals was from 31.3 to 76 days.The total number of ‘face-to-face’ follow-up visits by the geriatric psychiatrist and the assigned case manager for the patients on the active caseload of G-PAT was 1080.The total number of ‘solo’ follow-up visits by the case managers for the patients on the active caseload of G-PAT was 839.The total number of ‘telephone’ reviews by the geriatric psychiatrist and the case manager of the patients on the active caseload of G-PAT was 1179.

**Table 3 geriatrics-07-00118-t003:** The timelines for the initial assessment and first follow-up visits for 2019/2020.

Appointment Description	2019–2020												
	19-Apr	May	Jun	Jul	Aug	Sep	Oct	Nov	Dec	20-Jan	Feb	Mar	Total
The total number of new consults	13	15	22	17	21	19	21	20	9	22	23	11	213
Acceptance to first appointment “priority 1” (mean) in days	36.1	49.8	48.0	33.8	45.8	52.8	38.8	34.2	56.1	42.3	47.7	59.0	45.4
Acceptance to first appointment “priority 1” (median) in days	85.7	81.9	67.3	44.8	63.5	56.8	47.4	30.6	65.8	41.3	56.0	41.3	60.0
First follow-up appointment “priority 1”(mean) in days	76.0	31.3	39.3	50.6	31.3	61.1	44.3	35.3	22.3	35.3	18.4	61.2	42.2
Face-to-face follow-up visits the doctor and the case manager	110	102	89	88	89	87	84	89	78	74	71	119	1080
Solo face-to-face case manager visits	78	59	101	79	56	40	85	69	61	87	73	51	839
Telephone follow-ups by the doctor and/or case manager	85.0	49.0	92.0	89.0	57.0	53.0	104.0	84.0	72.0	162.0	131.0	201.0	1179.0
**Follow ups by the team (total)**													3098
The total active caseload of unique cases	38	37	41	36	45	43	45	37	47	45	54	57	525
ED visits for behavioral issues	1	0	1	0	0	1	0	0	0	0	1	0	4 (0.8%)
Active case load of G-PAT to BH from LTCH	1	1	1	1	0	1	0	0	1	0	1	0	7 (1.4%)

#### 3.3.3. Tertiary Outcome Measure

The total number of residents of LTCH in the City of Hamilton who visited the SJH emergency department, for all reasons (medical, surgical or behavioral), between December 2017 and June 2022 was 1943. Twenty-seven (27) patients, who were residing in long-term care homes (LTCH) in the city of Hamilton, presented to the psychiatric emergency services for the assessment of behavioral risks over a duration of 42 months. This averages to less than one patient every six weeks, who presented to PES for the assessment of behavioral risks. The total number of patients on the active case load of the G-PAT, who visited the emergency department for behavioral risks were 22. However, this number of 22 was over a duration of 36 months, versus the data from PES, SJH, which was over the duration of 42 months.

## 4. Discussion

The *Contemporaneous Model* of service delivery by the G-PAT at Hamilton Health Sciences (HHS), Hamilton, Ontario, has been structured after the success of the comprehensive geriatric assessment teams and tethered to the benchmarks put in place by RGP© and MOH, Ontario [4,13]. The results of the operations of G-PAT at HHS are the first of its kind, which focus on the impact of its operations in reducing the burden of the persons with dementia with behavioral risks on the emergency departments and the utilization of the specialized dementia behavioral services by this cohort of population

In general, there is a paucity of evaluative studies on the effectiveness of the operations of the geriatric mental health outreach teams in both non-institutional and institutional settings. For non-institutional settings, limited available data supports the role of this model in increased case finding in isolated seniors, and some degree of symptom reduction [7,8,9,10]. The data on the evaluation of the geriatric mental health outreach teams in institutional settings is even more limited. The outcome measures on the psychiatry lead multidisciplinary teams have shown an increase in the in-house staff’s capacity to independently manage residents’ needs [12]. The only published literature on the effectiveness of the geriatric mental health outreach team in preventing patients on its active caseload from visiting emergency departments is the one published by the senior author of this manuscript [13]. The current findings add to the body of evidence supporting the G-PAT modality in diverting emergency department visits. 

The variables identified from the meta-analysis of the C-GAT, which were responsible for their success, have also proven effective in the operations of the G-PAT in the community. The success of the G-PAT can be directly attributed to the principles governing the *Contemporaneous Model* of the service delivery. The *Contemporaneous Model* of service delivery takes complete ownership of the referral index problem, taking control of managing all of the variables deemed contributory to the presence of behavioral risks (psychiatric, medical, and psycho-social) and optimization of the use of psychotropic medications, offering very close surveillance and protracted follow-up, until the resolution of the index problems. To expand upon the results further, all level-1 priority referrals were seen in a consultation and care plan put in place within 30 days of the referral to the G-PAT. Additionally, the first follow-up for the level-1 priority patients was also within 30 days of the initial consultation for that patient. The strength of this team functioning is the ability to offer very close surveillance of the patients on the active caseload of the team. The total number of unique cases on the active caseload of G-PAT ranged from between 460 to 525, for each of the three years of the data collected. The number of ‘all’ follow-ups (face-to-face doctor and case manager, telephone review by doctor and case manager, and solo case manager) ranged from 3000 to 3200 for each of three years. This averaged approximately six (6) individual contacts for each patient on the active caseload of the G-PAT; obviously some patients may have been offered way more than six contacts in a given fiscal year, as this is always determined by the acuity of the situation. Facilities were responsible for identifying the patients with behavioral risks, which they felt were of higher acuity and with which they were unable to cope. This was in recognition of the fact that each individual care facility varies in their capacity to manage behavioral risks. It is often a continuum and ranges from care facilities, which are very well resourced and have the adequate skill set, to the ones who may not have the same level of resources and skill set. To that end, some of the patients may have received the bulk of their six follow-ups in the first 10 to 12 weeks of the initial contact, with the balance spread over the subsequent 12 weeks. It is an established scientific fact that it takes between four to six weeks to achieve optimal benefits after a change in dose of the commonly used psychotropic medication to manage behavioral expressions. Hence, an adequate surveillance routine should occur between every six to eight weeks to evaluate the effects of the changes in the pharmacological and behavioral care plan put in place. The G-PAT was able to deliver on this expectation, thereby ensuring it stayed on top of the monitoring and treatment of the index problems in all of the patients on its active caseload.

Additional reasons for the success of the *Contemporaneous Model* of service delivery of GOS was due to its evolution to meet the rapidly growing needs of the changing resident demographics of the LTCH [34]. It has done so by being inclusive of young adults with severe disabilities with co-morbid mental illnesses, aging residents with chronic mental illnesses with early-onset dementias, and multiple medical morbidities, in addition to the usual aging frail older persons with dementia and behavioral expressions. Hence, the inclusion criteria have shifted their focus to the presence of *Cognitively Complex care* needs of the residents in LTCH, *regardless of the age or etiology*, and in keeping with the changing demographics of LTCH. This is an essential and critical step, as almost all community-based clinical and social support services cease support of their *clients,* once admitted to LTCH.

Furthermore, the effective management of the behavioral risks in cognitively complex patients residing in nursing homes, and on the active caseload of G-PAT, significantly reduced the need for these patients’ referral to specialized dementia behavioral units for stabilization. This should enhance the capacity of the specialized behavioral units to accommodate cognitively complex patients with behavioral risks from the GIM units as well as other parts of the health care system. Enhancing the capacity of the health care systems has been proposed as one of the solutions to manage *Hallway Medicine* [35,36] and efficient operations of the G-PAT; working under the *Contemporaneous Model* of service delivery appears to be one such way to enhance the capacity of the system.

There are a few limitations in the interpretation of the outcome results in this study. Firstly, there is no control group, to which this data can be compared, thereby limiting the interpretation of the results. Secondly, the additional benefits of the G-PAT on the input offered by the in-house behavioral teams (BSO and PRC) in preventing visits to the emergency department or referral to specialized behavioral services, cannot be quantified in this study design. Finally, since the results of the project are descriptive and expressed as percentages, they are not amenable to deductive analysis, thereby limiting their interpretation.

## 5. Conclusions

Geriatric psychiatric assessment team, operating under the *Contemporaneous Model* of service delivery, appears to be an effective modality for reducing the burden of the older persons with dementia with behavioral risks on the emergency departments and the utilization of the specialized behavioral services. The role of G-PAT, governed by the *Contemporaneous Model* of service delivery, in diminishing the burden of *Hallway Medicine* needs further study. G-PAT, governed by the *Contemporaneous Model* of service delivery, has the potential to become the gold standard for the operations of the community-based geriatric mental health outreach teams across the province and the country.

## Figures and Tables

**Figure 1 geriatrics-07-00118-f001:**
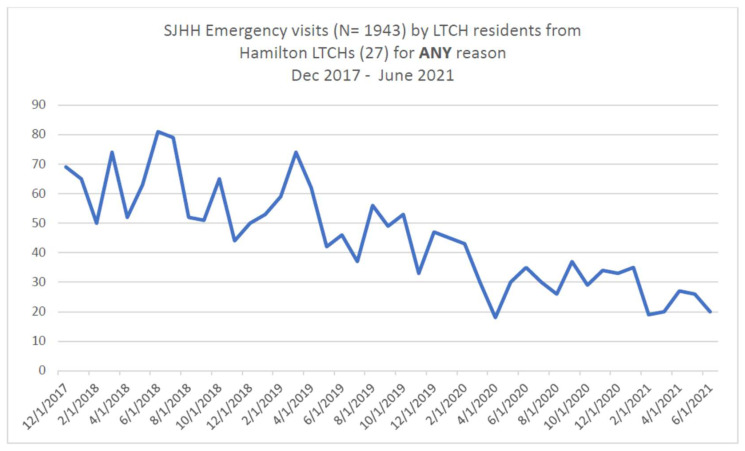
Shows the variability in the total number of residents of LTCH who visited the emergency department, for all reasons (medical, surgical or behavioral), at every two month interval, between December 2017 to June 2022.

## Data Availability

The authors confirm that the data supporting the findings of this study are available within the article and/or its supplementary materials.

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
