# Peer review of "Results of a Continuous Quality Improvement Initiative of the Contemporaneous Model of Service Delivery"

_geriatrics, 2022, doi:10.3390/geriatrics7050118_

Round 1

Reviewer 1 Report (Previous Reviewer 3)

Many thanks for the opportunity to review this paper.  I feel the change in format of the methods section now better displays the rigour of the approach and design, while the addition of the limitations section fully reflect the caveats necessary to apply the findings to practice.

The changes in language used are also welcome.

I'm happy for this paper to progress to being published.

Author Response

No changes to the manuscript were requested and the reviewer is pleased for the manuscript to move to publishing.  

Reviewer 2 Report (Previous Reviewer 2)

Dear authors,

Thank you for providing a point to point. Based on the document provided I think that the methods part have been improved significantly. However, to my view the introduction part needs to be shorter.

Kind regards,

Author Response

Reviewer 2.  This reviewer has requested that the “Introduction part needs to be shorter”.  I have significantly shortened the ‘Introduction’ section in keeping with the request made. 

Please review and let me know if anything else is required from my end. 

Sincerely 

A.S. Luthra MD MSc FRCFC

This manuscript is a resubmission of an earlier submission. The following is a list of the peer review reports and author responses from that submission.

Round 1

Reviewer 1 Report

Thank you for the opportunity to review your paper - Contemporaneous Model of Service delivery for managing behaviors in residents of long-term care homes; Results of a 3 year 3 Continuous Quality Improvement Initiative.

Overall, the content of this manuscript is interesting and the topic showed originality. However, there are areas/sections in which the authors should consider revising:

TITLE: The title is too long and should be revised.

ABSTRACT: In the abstract, the objective is not clearly defined, as well as the method is not explained.

INTRODUCTION: introduction provide sufficient background and include all relevant references, but it lacks to include the objective of the article.

MATERIALS AND METHODS: The methodology used in the study is not clear. The inclusion and exclusion criteria for defining the participants is not clear. The number of the ethical clearance of the ethics committee was not presented

RESULTS: the results are clearly presented

CONCLUSION: As the objectives of the article are not clearly defined, it is not clear whether the conclusion responds to the objectives of the article. I suggest clarifying the objectives and methodology to understand the conclusions.

Author Response

  1. The introduction part of the manuscript has been reformatted to clearly articulate the purpose and the aim of this manuscript such that everything else seems to follow suit.
  2. The other raised issue  was the need to improve the ‘Materials and Methods’ section of the paper. The section has been rewritten to incorporate the feedback around the clarification of the inclusion and exclusion criteria, the ethical clearance issues, the role of the case managers and the articulating the process followed by the team, from the time of referral to the time of seeing the patient. 
  3. The title is too long and this has been changed. The abstract has been rewritten to define the objective of the study, more clearly.  Also, the objectives and the methodology section has been rewritten such that the conclusions are more contextual.

Reviewer 2 Report

Thank you for giving me the chance to review this interesting article. Please find below my comments:

-        The abstract is very unclear. Which is the aim, which are the key results? Please reformulate it.

-         Row 28 a reference is needed;

-        You say about a meta-analyses at row 28 but you are giving 4 references at the end of the sentence;

-        Rows 36-40. This is a very long sentence. Please split;

-        The role of case managers is not very clear? What exactly they do etc.;

-        Rows 54-55. Please cite the year.

-        Rows 75-80. To my view you can delete them;

-        Rows 81-104 should be presented with less text. Too much information that is not needed;

-        The way it is presented what the paper will present is more like a report rather than a scientific article; Please correct;

-        I suggest presenting in a very clear way which is the aim of the study;

Methods

-        This part is very short. A detailed information is needed.

Results

-        The result section is just descriptive. This doesn’t give much info about the usability of the results;

Discussion/Conclusions

-        A section about strengths and limitations is missing;

-        Some parts of the discussion section presents again the results. Please avoid repetition;

Author Response

  1. The introduction part of the manuscript has been reformatted to clearly articulate the purpose and the aim of this manuscript such that everything else seems to follow suit.
  2. The other raised issue was the need to improve the ‘Materials and Methods’ section of the paper. The section has been rewritten to incorporate the feedback around the clarification of the inclusion and exclusion criteria, the ethical clearance issues, the role of the case managers and the articulating the process followed by the team, from the time of referral to the time of seeing the patient. 
  3. Additional references of the meta-analysis has been added to row 28.  Rows 36-40 have been split into two sentences.  Row 54-55, I have added the year. Rows 75-80 has not been deleted but modified (reviewer 3 comments).  I respectfully disagree with the comment on the rows 81-104.  It is required as it defines the new framework for this model.  Methods section has been expanded.  A section on limitations has been added to the discussion. Repetition of the results in the discussion section has been removed. By clarifying the objectives, material and methods section, the manuscript reads as an outcomes evaluation study.  

Reviewer 3 Report

Hi - thank you for the opportunity to review this paper which offers some interesting insights in the field.  However, at the moment I feel there is limited clarity in terms of methodology and approach, which makes it difficult to appraise its rigour. This must be addressed before publication.

Please see the comments below for specific feedback:

·        Some typographical errors and repetition of phrased – a final proof read is needed before acceptance.

·        Lines 75-80: I feel this paragraph is too specific to the setting of this paper.  Consider revising to illustrate general and transferable issues to readers elsewhere.

·        Lines 116-145: I feel the material and methods section requires considerable development, as this information is largely either process or definition-driven.  Some reflection on overall study design, data collection, data analysis, ethical considerations and quality assurance processes could be included here to allow appraisal of the study’s methods, rather than reflection on the results alone.

·        Are there demographic details about the study population which can be added?  Without these it is difficult to monitor your findings for extraneous variables.

·        Results section: rather than being structured around individual years, which makes changes in the outcomes difficult, I would like to see this section structured around the outcome measures themselves over time.  This is difficult for the reader to follow otherwise.

·        Lines 247-250: is there a comparison point for this data to illustrate any change?  It is difficult to confirm effectiveness without this.

·        Lines 281-282: I feel there is some conjecture in this statement – what evidence do you have for saying the success is directly attributable to these principles?  This was not what you investigated.

Author Response

  1. The introduction part of the manuscript has been reformatted to clearly articulate the purpose and the aim of this manuscript such that everything else seems to follow suit.
  2. The other raised issue was the need to improve the ‘Materials and Methods’ section of the paper. The section has been rewritten to incorporate the feedback around the clarification of the inclusion and exclusion criteria, the ethical clearance issues, the role of the case managers and the articulating the process followed by the team, from the time of referral to the time of seeing the patient. 
  3. Typographical errors and repetition has been corrected. Lines 75-80 have been modified.  Lines 116-145 have been reformatted to include the feedback.  The changing and the heterogeneous population of LTCH has been explained and this will be the case, moving forward.  In my opinion, including demographics does not add any more value to the findings of the paper.  I need to keep the results section separate for each of the three years.  I have added to the results for them to offer more clarity.  The comments on the lines 247-250 have been addresses by adding the ‘limitations’ section.  Lines 281-282 have been addressed by adding the ‘limitation’ section and by better formatting the objectives of the study. 

Round 2

Reviewer 2 Report

Dear authors,

Thank you for addressing the comments. Most of them have been addressed accordingly. However, I have two more comments that think should be addressed.

- The methods part is still very poor. To my view, more info is needed.

- The results part is just descriptive. If not possible for a deductive analysis, please mention this as a limit of the current work. 

Reviewer 3 Report

Many thanks for the opportunity to review this paper and its amendments, several of which help the reader follow the study with a lot more clarity.

However, I still feel the material and methods section reflects the process and outcomes of the model, rather than that of a research paper. Elsewhere in the journal, the common subheadings commonly follow structure such as design, participants / sample, intervention, measurement, procedure and analysis (or variants thereof). These allow the reader to appraise how the knolwedge was created (rather than how the model was delivered) to ensure scrutiny and rigour (as to facilitate a CASP appraisal for example). Without this, it is difficult to judge the standard of the paper.

This would also help facilitate more comparison to existing knowledge beyond the study setting alone, which is missing from the conclusion section.

Can I also recommend the following article for the use of non-ageist language in research literature, as I note there are instances when older people are referred to as a burden / a focus on the needs of services and not the population itself?

Carmen Bowman & Weng Marc Lim (2021) How to Avoid Ageist Language in Aging Research? An Overview and Guidelines, Activities, Adaptation & Aging, 45:4, 269-275, DOI: 10.1080/01924788.2021.1992712